# Superinfection exclusion creates spatially distinct influenza virus populations

**Anna Sims[1]\***, **Laura Burgess Tornaletti[1]**, **Seema Jasim[1]**, **Chiara Pirillo[2]**, **Ryan Devlin[2]**, **Jack C. Hirst[1]**, **Colin Loney[1]**, **Joanna Wojtus[1]**, **Elizabeth Sloan[1]**, **Luke Thorley[1]**, **Chris Boutell[1]**, **Edward Roberts[2]**, **Edward Hutchinson[1]\***

**1** MRC-University of Glasgow Centre for Virus Research, Glasgow, United Kingdom, **2** Beatson Institute for Cancer Research, Glasgow, United Kingdom

\* a.sims.1@research.gla.ac.uk (AS); Edward.Hutchinson@glasgow.ac.uk (EH)

## Abstract

Interactions between viruses during coinfections can influence viral fitness and population diversity, as seen in the generation of reassortant pandemic influenza A virus (IAV) strains. However, opportunities for interactions between closely related viruses are limited by a process known as superinfection exclusion (SIE), which blocks coinfection shortly after primary infection. Using IAVs, we asked whether SIE, an effect which occurs at the level of individual cells, could limit interactions between populations of viruses as they spread across multiple cells within a host. To address this, we first measured the kinetics of SIE in individual cells by infecting them sequentially with 2 isogenic IAVs, each encoding a different fluorophore. By varying the interval between addition of the 2 IAVs, we showed that early in infection SIE does not prevent coinfection, but that after this initial lag phase the potential for coinfection decreases exponentially. We then asked how the kinetics of SIE onset controlled coinfections as IAVs spread asynchronously across monolayers of cells. We observed that viruses at individual coinfected foci continued to coinfect cells as they spread, because all new infections were of cells that had not yet established SIE. In contrast, viruses spreading towards each other from separately infected foci could only establish minimal regions of coinfection before reaching cells where coinfection was blocked. This created a pattern of separate foci of infection, which was recapitulated in the lungs of infected mice, and which is likely to be applicable to many other viruses that induce SIE. We conclude that the kinetics of SIE onset segregate spreading viral infections into discrete regions, within which interactions between virus populations can occur freely, and between which they are blocked.

## Introduction

In order for viral genomes to directly interact, viruses must infect the same cell. Viral genome interactions during coinfection can shape viral fitness, population diversity, and adaptation. The most dramatic examples of this include the recombination events between SARS-related coronaviruses that contributed to the emergence of the Severe Acute Respiratory Syndrome Coronavirus 2 (SARS-CoV-2) [1,2], and the repeated generation of influenza A virus (IAV) pandemics by reassortment between different strains of the virus [3,4]. In the case of IAVs, coinfection also provides

**Data Availability Statement:** All relevant data are within the paper and its Supporting Information files. However, we will also make primary data available through our Institutional data repository at http://dx.doi.org/10.5525/gla.researchdata.1370.

**Funding:** We acknowledge funding from the UK Medical Research Council (MRC), as studentships to A.S., J.H. and J.W. [MC_ST_CVR_2019], as CVR core funding to C.B. [MC_UU_12014/5] and as a Career Development Award and Transition Support Award to E.H. [MR/N008618/1 and MR/V035789/1]; funding from the University of Glasgow to E.H., and funding from Cancer Research UK (CRUK) to E.R. [A_BICR_1920_Roberts]. The funders had no role in study design, data collection and analysis, decision to publish, or preparation of the manuscript.

**Competing interests:** The authors have declared that no competing interests exist.

**Abbreviations:** BSA, bovine serum albumin; DMEM, Dulbecco's Modified Eagle Medium; FFU, focus forming unit; GFU, green forming unit; HEK, human embryonic kidney; IAV, influenza A virus; MDCK, Madin–Darby canine kidney; MOI, multiplicity of infection; PFU, plaque-forming unit; RFU, red forming unit; SARS-CoV-2, Severe Acute Respiratory Syndrome Coronavirus 2; SIE, superinfection exclusion; SST, total sum of squares; VGM, viral growth media; WT, wild-type.

opportunities to modulate gene expression, to increase fitness by restoring error-free genomes, and to decrease fitness through competition with defective-interfering genomes [5–7].

In the case of IAVs, multiple lines of evidence show that coinfection is possible. Coinfection of cells with IAV has been demonstrated on many occasions in experimental settings [3,8–13]. Under the right circumstances, reassortment in cell culture models of infection can be frequent and over a single round of infection the proportion of reassortants increases exponentially with the frequency of coinfection [8]. Coinfection can also be demonstrated experimentally in vivo: for example, when using a virus that was dependent on coinfection for replication, amplification of the virus was observed in the nasal passages of guinea pigs, indicating that coinfection occurred during multiple rounds of infection [9]. Coinfections can also be inferred from observations of natural infections, such the emergence of reassortant IAV strains in humans [14] and the frequent detection of reassortant IAV genomes in wild birds [15–18]. These data demonstrate that coinfection of cells is possible, but they do not provide information about how frequently coinfection occurs in natural infections. Because IAVs replicate rapidly, a reassortant strain with a fitness advantage could rapidly become very common even if the coinfection event that generated it was rare. As a result, although coinfection clearly can occur during IAV infections, the how likely it is to occur in any given cell is unclear.

Outside of an experimental setting, it is unlikely that 2 unrelated virus particles would reach the same cell at exactly the same moment. Instead, one would expect unrelated viruses to replicate locally within a host organism before eventually encountering the same cell. For this reason, we assume that coinfection most commonly occurs by superinfection: the infection of a previously infected cell. For many viruses, the potential for superinfection is strongly limited, as following the initial infection changes occur within a cell that block its permissivity to secondary infection. This phenomenon is known as superinfection exclusion (SIE) and has been described for many viruses of bacteria, animals, and plants [19–24]. SIE can occur through a wide range of mechanisms, but it occurs at the level of individual cells and involves the exclusion of closely related viruses—in this way, it is distinct from viral interference, where the replication of 1 type of virus in a host suppresses the replication of another [25]. For SIE, the amount of time required for a cell to become resistant to secondary infection varies depending on virus and cell type [19,20,26]. SIE has been described for IAV, and for laboratory-adapted strains of IAV in monolayers of transformed cells the interval between primary and secondary infection required for robust SIE has been reported to be around 6 h [27,28].

Studies of SIE and cellular coinfection often infer coinfection from the genotypes of the reassortant viruses produced during an infection and lack spatial information on how SIE controls the potential for coinfection within hosts [29]. Recent studies of IAV in animal models demonstrated that anatomical compartmentalisation can restrict opportunities for coinfection, as viruses that infect different sites in the respiratory tract or even different locations within the lung have limited opportunities to reassort [12,30,31]. In the respiratory tracts of human patients and of experimentally infected animals, multiple viruses including IAV form discrete foci of infection, indicating that the viruses are able to spread to directly adjacent cells as they propagate within the airway [32–37]. As these foci of infection grow and meet, viruses that entered the host separately would have the opportunity to coinfect cells. Because SIE should limit interactions between these foci, we reasoned that it could be an important factor in controlling the rates of coinfection and reassortment [38]. We therefore wished to investigate potential for coinfection when viruses propagate over multiple rounds of infection and spread through multiple cells. To do this, we needed to visualise infected lesions directly, which we did using fluorescent reporter viruses, an approach that has been used previously to identify coinfected cells in vitro, and which has been used to identify coinfected cells in vivo following high-dose intranasal inoculation of mice [39,40].

In this study, we investigated the effects of SIE on interactions between spreading infections using "ColorFlu," a system of isogenic reporter viruses developed from the laboratory strain A/Puerto Rico/8/1934 (H1N1; PR8), which differ only in the nature of a fluorophore tag fused to the viral NS1 protein [39,41]. We first used ColorFlu in individual cells to define the kinetics of SIE induction. By varying the interval between adding viruses encoding different fluorophores, we showed that SIE is not effective for the first 2 h of infection, but that after this point, the effects of SIE increase exponentially with time. We found that the primary virus establishes a robust barrier to superinfection that cannot be readily overcome by increasing the amount of superinfecting virus. We then asked how the onset of SIE would constrain interactions between virus populations as IAV spread locally through multiple cells. Using a cell culture model, we found that the kinetics of SIE onset had 2 distinct effects as viruses spread across multiple cells. Within a single focus of infection, SIE does not restrict coinfection between progeny viruses as the plaque expands. However, when 2 separate and growing infected regions meet, SIE restricts coinfection between their virus populations. This creates a pattern of discrete virus subpopulations, which was recapitulated in lesions in the lungs of infected mice. As this patterning is dependent on the timing of SIE onset, rather than a specific mechanism, we expect that we would observe the same phenomenon for a range of viruses. Our data show that SIE defines the size of the regions where coinfection between related viruses can occur within a host, and hence controls viral fitness and evolution.

## Results

### Coinfection between influenza A virus is initially unrestricted and then reduces exponentially

We wanted to quantify the effect of SIE on coinfection between IAVs without having to account for potential functional incompatibilities between different strains. We therefore used isogenic reporter IAVs (ColorFlu), which differ only in the fluorophore they encode [39]. In segment 8 of their genome, these viruses encode a fluorophore (in this study, either mCherry or eGFP) as a C-terminal fusion to the NS1 protein. As shown in Fig 1A, Madin–Darby canine kidney carcinoma (MDCK) cells infected with these viruses appear either green (if infected only with eGFP viruses; green in figures), red (if infected only with mCherry viruses; magenta in figures), or yellow (if coinfected with both variants; white in figures). The 2 viruses used had no significant differences in growth when assessed by single cycle (S1A Fig) and multicycle (S1B Fig) growth kinetics (Mann–Whitney U test, $p > 0.05$).

Using this system, we wished to measure kinetics of the reduction in superinfection potential due to the onset of SIE. To do this, we infected MDCK cells with eGFP-tagged (green) virus and then with mCherry-tagged (red) virus, both at a multiplicity of infection (MOI) of 1 FFU/cell, varying the time interval between the 2 infections. We harvested the cells at 16 h after the secondary infection and measured the expression of both fluorophores by flow cytometry (Fig 1B). Initially, coinfection between the viruses was unrestricted, and no significant change in the proportion of cells expressing mCherry was detected for intervals between infections of 1 or 2 h. However, as the interval between infections increased, the cells became less permissive to secondary infection and the proportion of cells able to express mCherry decreased (Fig 1C), with a significant reduction in the percentage of coinfected cells observed with an interval of 3 h ($p = 0.0074$, Kruskal–Wallis test) and at every subsequent interval ($p < 0.0001$). The percentage of cells expressing mCherry declined to nearly zero once the interval between infections reached 7 h (Fig 1C). Our data are consistent with previous studies that detected robust SIE when there is an interval of 6 h between infection events [27,28]. Additionally, we show that up to 2 h post primary infection there is no restriction on

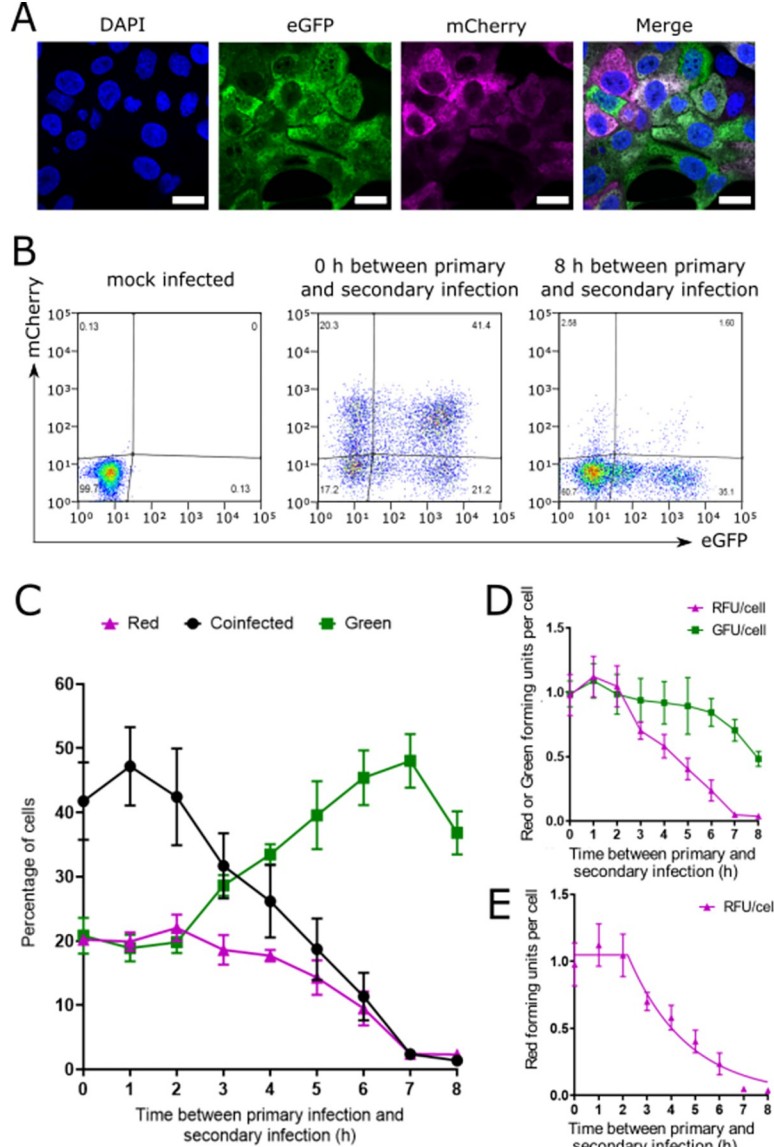

**Fig 1. The ability of IAV to cause SIE depends on the interval between primary and secondary infections.** (A) Confocal micrographs of cells infected with ColorFlu reporter viruses, which cause cells to express eGFP (green) or mCherry (magenta). Co-expression of both fluorophores is shown as white. MDCK cells were grown on glass coverslips, infected at an MOI of 1 FFU/cell for each virus, and fixed at 8 h post infection (hpi). Images were obtained using a 64× objective. Scale bar = 2 μm. (B) Flow cytometry of cells infected with reporter viruses. MDCK cells were first infected with ColorFlu-eGFP, before secondary infection at the time points indicated with ColorFlu-mCherry, with both viruses at MOI 1 FFU/cell. Representative plots are shown. (C) Kinetics of onset of SIE, determined from flow cytometry analysis; means and SD are shown ($n = 6$). Differences in the percentage of coinfected cells, compared to simultaneous infection (time = 0 h), were tested for significance by one-way ANOVA. By 3 h, the difference was significant ($p = 0.0074$), and at every subsequent time point ($p < 0.0001$). (D) The number of reporter viruses per cell that were able to cause expression of their fluorophore, with varying intervals between infection with primary (green) and secondary (red) viruses. Viruses were quantified as GFUs and RFUs, calculated from the proportions of red, green, and coinfected cells. The mean and SD are shown ($n = 6$). (E) The relationship between the expression of the secondary virus and the interval between infections, as shown in (D), modelled as an initial period of no SIE followed by an exponential increase in SIE. SST = 0.74. Underlying data can be accessed at the following address: http://dx.doi.org/10.5525/gla.researchdata.1370. FFU, focus forming unit; GFU, green forming unit; IAV, influenza A virus; MDCK, Madin–Darby canine kidney; MOI, multiplicity of infection; RFU, red forming unit; SIE, superinfection exclusion; SST, total sum of squares.

coinfection, and thereafter, there is a progressive shift in the cells from a permissive to an exclusionary state as intervals between infections increase.

To examine the effects of SIE more clearly, we considered gene expression from the primary (eGFP) and secondary (mCherry) viruses separately. To do this, we used the proportions of cells expressing one or both fluorophores to infer the numbers of viruses per cell that had caused expression of eGFP (green forming units, GFUs) or mCherry (red forming units, RFUs). This was done by assuming that viruses administered at the same time infected cells independently of each other and that infection could therefore be modelled by a Poisson distribution (see Materials and methods for details). We found that the number of GFU per cell (primary virus) remained consistent as the interval between infections was increased, up to 6 h. With intervals of greater than 6 h, we observed a reduction in GFU per cell, which we attributed to infected cells to becoming detached and lost from the analysis at late time points. This was consistent with a proportional increase in the detection of uninfected cells when there was an interval of 7 h or 8 h before secondary infection (S2A Fig). Conversely, although the number of RFU per cell (secondary virus) remained consistent for intervals of up to 2 h, after this point it declined, demonstrating the onset of SIE (Fig 1D).

The mechanism for SIE in IAVs is not yet known, but it has previously been suggested that it may require an actively replicating influenza polymerase [28]. As the products of viral transcription and replication appear to accumulate exponentially in a newly infected cell after an initial lag for cell entry, uncoating and transport of the viral genome to the nucleus [42,43], we reasoned that the inhibitory factor that drives SIE might also increase exponentially once a primary infection is established. If this were the case, we would expect the RFU per cell to fit a model in which there was initially no SIE but where, after the interval between infections increased beyond a certain point, SIE increased exponentially. To test this hypothesis, we fitted a model of the RFU per cell at different intervals between infections, in which a period of constant expression of RFU was followed by an exponential decay (Fig 1E). This model was a good fit to the experimental data (total sum of squares (SST) = 0.74), with an initial constant phase of 2.2 h (95% CI 1.8 to 2.6 h) followed by an exponential decay with a half-life of 1.7 h (95% CI 1.4 to 2.1 h). We noted that the 7 h and 8 h samples do not fit the model as closely as the other points, which we again attributed to death of infected cells. In separate experiments, we found that this model describes the kinetics of SIE whichever virus was used for the primary infection (S2B and S2C Fig, SST is 0.22 and 0.38, respectively), although the parameters of the model differed depending on which virus was used first, suggesting that the kinetics of SIE may be modulated by the reporter fluorophore. Together, these data suggest that IAV SIE can be explained by an inhibitory factor, which is initially absent but then accumulates exponentially after infection with similar kinetics to IAV gene expression.

## The kinetics of superinfection exclusion is impacted by the dose of primary infecting virus, but not the secondary infecting virus

Once we could model the kinetics of SIE onset, we were able to ask how this changed in response to the conditions of infection. To compare the change in kinetics between conditions, we measured the RFU per cell and used this to fit the model above, constraining the initial constant phase to 2 h and the plateau of the decay phase to 0, and then compared the half-life of the decay phase. We removed the confounding effects of cell death by only considering intervals between primary and secondary infection up to 6 h (S2A Fig).

We began with equal ratios of primary and secondary viruses (both at an MOI of 1 FFU/cell). Under these conditions, half-life of the decay phase was 2.3 h (Fig 2A). We then increased the amount of primary virus (Fig 2A, upper panels) or secondary virus (Fig 2A, lower panels)

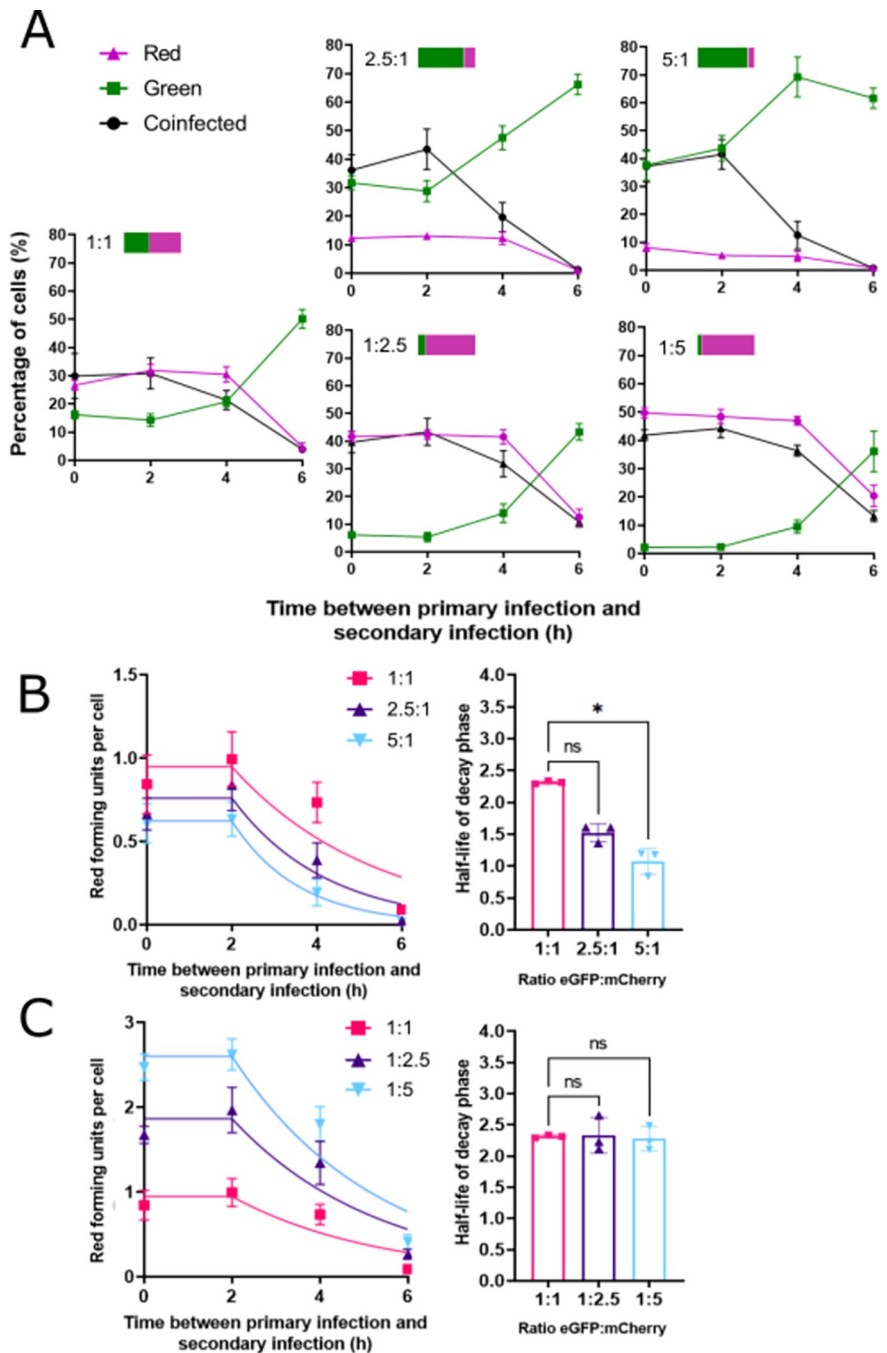

**Fig 2. SIE kinetics are sensitive to the amount of primary infecting virus. (A)** To assess the effects of altering the ratios of primary (ColorFlu-eGFP, green) and secondary (ColorFlu-mCherry, magenta) infecting viruses, SIE was measured as in Fig 1. Five different conditions are shown, with the ratios of primary and secondary viruses for each experiment indicated as bars. The initial FFU/cell for 1:1 ratio is 0.66 for ColorFlu-eGFP and 0.72 for ColorFlu-mCherry. These data were used to calculate how the expression of secondary virus (RFU per cell) changes with the interval between infections. This is shown when changing the ratios of **(B)** primary and **(C)** secondary infecting viruses. The RFU per cell was then fit to a model describing an initial constant phase of 2 h, followed by exponential decay plateauing at 0 (B, C: left-hand panels). The SST of the models fitted for 1:1, 2.5:1, and 5:1 are 0.43, 0.18, and 0.067, respectively. The SSTs for the models fitted for 1:1, 1:2.5, and 1:5 are 0.43, 1.01, and 1.10, respectively. The half-life of the decay phase, after the initial constant phase of 2 h, was then calculated (B, C: right-hand panels). Differences between these intervals and those observed with a 1:1 ratio were determined by Kruskal–Wallis test (*$p < 0.05$). For all data the mean and SD are shown ($n$ = 3). Underlying data can be accessed at the following address: http://dx.doi.org/10.5525/gla.researchdata.1370. FFU, focus forming unit; RFU, red forming unit; SIE, superinfection exclusion; SST, total sum of squares.

by 2.5-fold or 5-fold. The model was a good fit to the data under all conditions tested (SST $\leq$ 0.46 when increasing the primary virus and $\leq$ 1.1 when increasing the secondary virus). Using the model, we compared the predicted half-life of the decay phase and found that SIE onset was more rapid when the amount of primary virus increased (Fig 2B; for a 5-fold increase in primary virus the half-life differs significantly from its value when equal input was used, $p < 0.05$ by Kruskal–Wallis test) but was not significantly altered by comparable changes in the amount of secondary virus (Fig 2C). This indicates that the kinetics of SIE are sensitive to the amount of primary infecting virus, and that SIE cannot be outcompeted by comparable variations in the secondary infecting virus. This suggests that, once established, the exclusionary state in the cells established by SIE would be hard to overcome.

## Superinfection exclusion does not restrict interactions within a spreading influenza A virus infection

Once we had defined the kinetics of SIE in individual cells infected with IAV, we wanted to investigate how SIE controlled the spread of IAV across multiple cells. We first asked whether SIE prevents the progeny viruses of an infection from interacting with each other as the infected focus expands. To examine this, we set up plaque assays, which allow viruses to propagate through MDCK cells under agarose, as a simplified model of the foci of infection observed in infected patients [34]. To study interactions between the progeny of a single infected cell, we established a system that in which we could infect individual cells with a mixture of red and green viruses. To do this, we first infected MDCK cells with a mixture of green and red viruses, both at a high MOI of 5 plaque-forming units (PFU)/cell, to create a population of coinfected cells. At 1 h post infection, before new virus particles were produced, we dispersed these infected cells using trypsin, diluted them, applied them to a fresh MDCK monolayer, and overlaid with agarose. This meant that each coinfected cell applied to the monolayer would be an individual PFU, shedding both red and green viruses into the same region (Fig 3A).

We proposed 2 hypotheses for how the viral progeny of these cells might interact to produce plaques. Either (i) rapid SIE onset would inhibit coinfection, resulting in the initial yellow focus segregating into discrete regions where 1 fluorophore would dominate; or (ii) SIE would not develop quickly enough to prevent coinfections at the plaque edge, and so the plaque would remain coinfected as it grew (Fig 3B). We observed that as coinfected plaques expanded, both fluorophores were expressed across the entire plaque area (Fig 3C). Therefore, we concluded that the cells at the leading edge of the plaque were receiving multiple viruses quickly enough for coinfection to occur before the effective onset of SIE.

Although both mCherry and eGFP expression could be detected across the plaques, we did notice that coinfected cells were concentrated towards the middle of the plaque area (Fig 3D). We quantified this by measuring the areas of coinfected regions and comparing this to the total infected area. We found the coinfected portion of plaques is at its highest at 24 hpi and is significantly reduced in the larger plaques seen at 48 and 72 hpi (Fig 3E, $p < 0.0001$ in both cases). However, this change in the distribution of fluorescent cells may not be due to changes in SIE, as live-cell imaging showed infected cells migrating to the centre of plaques, possibly as they began to die and detach from the substrate (S1 Movie). Taking our data together, we concluded that within a focus of infection, the kinetics of SIE allow coinfection to occur freely.

## Superinfection exclusion strongly inhibits interactions between established regions of influenza A virus infection

Next, we wanted to assess whether coinfection was restricted when 2 separate foci of infection expand and interact with each other. This provides a simple model of a natural infection,

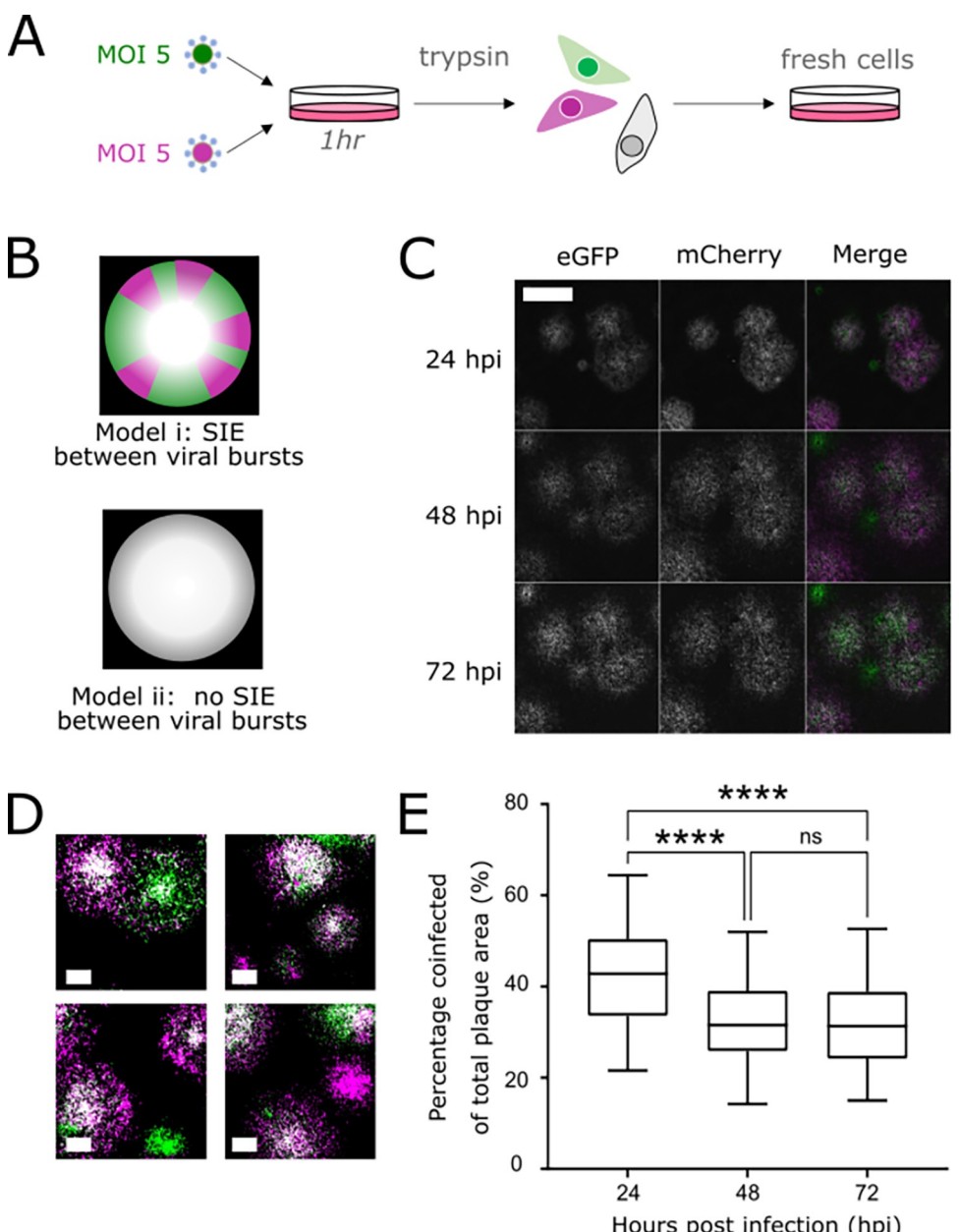

**Fig 3. SIE does not inhibit coinfection between IAVs from a single focus of infection. (A)** Experimental design for investigating the role of SIE in the spread of coinfected foci. **(B)** Proposed models for the spread of coinfected foci. **(C)** The spread of coinfected plaques, showing the same region at 3 different time points. Viruses were seeded onto monolayers of MDCK cells, overlayed with agarose, and imaged every 24 h. Images were taken on Celigo fluorescence microscope and a representative field of view is shown. Scale bar = 2 mm. **(D)** A binary threshold was applied to images of plaques to distinguish coinfected cells (white) from singly infected cells (magenta or green); representative images of plaques at 48 hpi are shown. Scale bars = 1 mm. **(E)** The percentage of total plaque area that was coinfected, calculated from binarised images at taken at each time point. Box and whisker plots show the percentages of infected areas from 71 individual fields of view at 3 time points in 1 experiment. Differences between the coinfected percentage at different time points were tested for significance by one-way ANOVA (**** $p < 0.0001$). Underlying data can be accessed at the following address: http://dx.doi.org/10.5525/gla.researchdata.1370. IAV, influenza A virus; MDCK, Madin–Darby canine kidney; SIE, superinfection exclusion.

where is unlikely that multiple viruses would enter a host organism and reach the same cell within a short space of time. Instead, we assume that within a host coinfection of cells by different strains of virus typically occurs through interactions between the progeny of separate foci of infection. To model interactions between spreading foci of infection, we infected MDCK monolayers at a low MOI with green and red viruses, overlaid them with agarose, and imaged the spread of plaques every 24 h for 72 h. We observed that, as adjacent plaques expressing different fluorophores grew towards each other, regions of cells expressing different fluorophores remained almost entirely distinct (Figs 4A and S3). On close examination, we observed a very thin boundary region of cells in which both fluorophores were expressed (Fig 4B). Image analysis showed that the coinfected region at 72 hpi was only around 1% of the total plaque area (Fig 4C). This indicates that only a small region of coinfection was possible before further interactions were blocked by the onset of SIE.

To investigate whether this exclusion phenotype was relevant to infections in vivo, we performed a version of this experiment in which IAV spread through the lung of a mouse. To do this, we infected C57BL/6 mice intranasally with a mixture of ColorFlu-eGFP and ColorFlu-mCherry (500 PFU of each virus), took sections of lungs from mice at day 3 or 6 post infection, and looked for regions where red and green foci of infection were interacting. Images of lungs harvested at day 3 suggested that the initial sites of infection were mainly in the bronchi (S4 Fig). However, by day 6 infection had spread into the alveoli and established distinct red and green lesions. Consistent with previous reports [39], coinfection was visible at many sites, which presumably received a high dose of both viruses simultaneously. However, at this time point, we also observed multiple instances where red and green lesions were adjacent to each other but still maintained only a small area where coinfection was supported, despite a lack of obvious anatomical compartmentalisation (Figs 4D and S5). This reflects the phenotype we observed in cell culture and suggests that SIE spatially restricts interactions between virus populations within a host organism. We therefore concluded that when 2 distinct regions of infection meet each other, SIE inhibits coinfection and the opportunities for reassortment between virus populations that it brings.

## Discussion

SIE has been observed for many different viruses of plants, bacteria, and animals [19–24] and, despite its implications for the evolution of medically important viruses such as the influenza viruses, it is not well understood. SIE constrains the ability of related viruses to asynchronously coinfect cells, as happens when viruses replicate and spread locally within a host organism. Studying the spread of animal viruses within their hosts is often challenging due to the difficulties in accessing sites of infection within a living animal [29]. Here, we used a simplified cell culture model of infection with isogenic, fluorescently tagged viruses to demonstrate how the kinetics of SIE onset limit coinfection during spreading IAV infections. We show that during the local spread of IAV infection from cell to cell, SIE defines the regions where coinfection can and cannot occur. The segregation of viruses into distinct regions that we observed was recapitulated in vivo when the tagged viruses were allowed to spread in the lungs of mice.

Our data imply that the kinetics of SIE onset control opportunities for viral genomes to interact in a spreading infection. They mean the progeny of a single virus can undergo sustained interactions within an infected focus, within which it would benefit from interactions that allow complementation between related genomes. For IAV, genome complementation within an infected lesion would mean that the semi-infectious progeny virions that make up around 90% the viral population could contribute to productive infection [44–46]. In a similar way, stochastic simulations of the effects of SIE in bacteriophage have demonstrated that viral

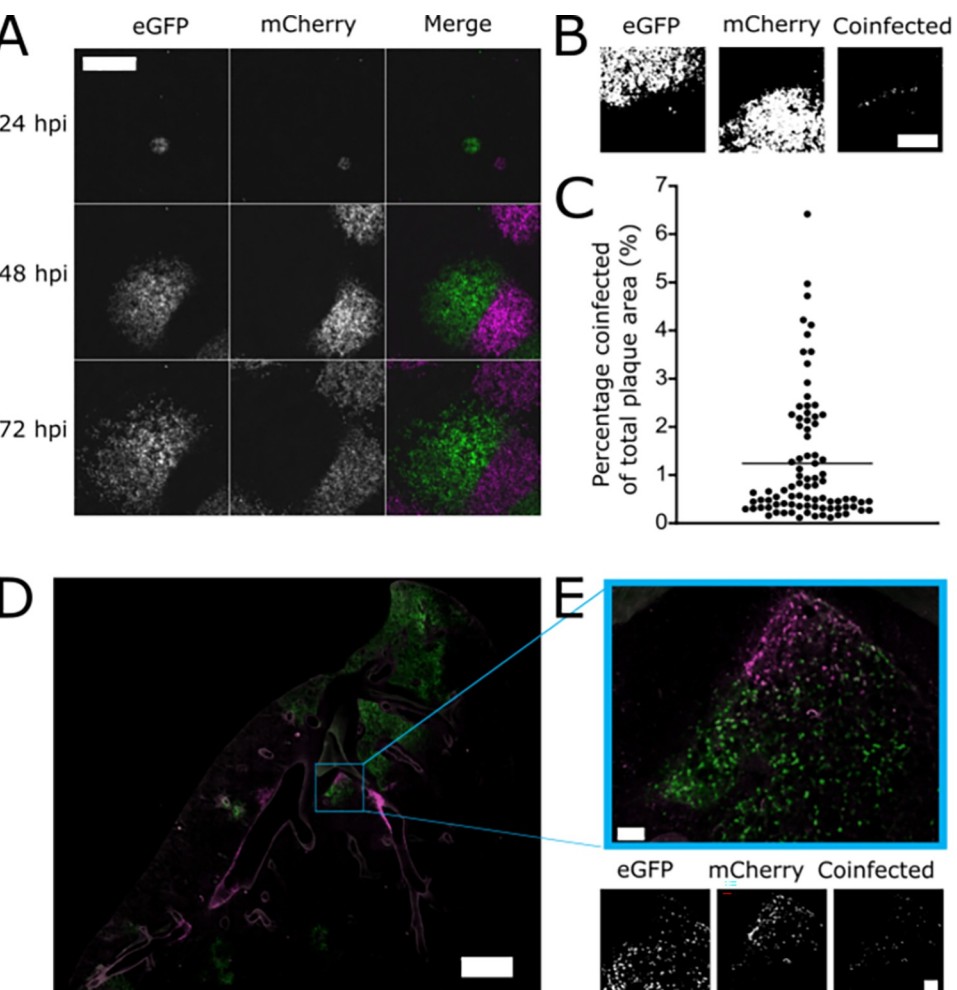

**Fig 4. SIE allows only a small region of coinfection when 2 established regions of IAV infection meet. (A)** Representative image of plaque interaction. Reporter viruses were seeded onto monolayers of MDCK cells, overlayed with agarose, and imaged every 24 h. Representative images, taken on Celigo fluorescent microscope, are shown. Scale bar = 2 mm. **(B)** A binary threshold was applied to images of plaques to distinguish cells expressing the eGFP, mCherry, or both fluorophores together. Images of a representative plaque are shown. Scale bar = 1 mm. **(C)** The percentage of coinfected areas in comparison to total plaque area was calculated from images at taken at 72 hpi. The mean and the percentage areas of 86 individual fields of view from 1 experiment are shown. **(D)** Lung sections from infected mice at 6 dpi. C57BL/6 mice were intranasally inoculated with mixtures of mCherry and eGFP expressing viruses (500 PFU of each virus). Lungs were harvested, sectioned, and imaged with an LSM 880 confocal microscope (Zeiss) using a 20× objective; scale bar = 1,500 μm. **(E)** Enlarged image of a lesion showing coinfection; scale bars = 100 μm. Binarised images of the lesion were produced as described in (B). Underlying data can be accessed at the following address: http://dx.doi.org/10.5525/gla.researchdata.1370. dpi, days post infection; IAV, influenza A virus; MDCK, Madin–Darby canine kidney; PFU, plaque-forming unit; SIE, superinfection exclusion.

populations that are incapable of initiating SIE would be less able to fix beneficial mutations [47]. We suggest that for IAV coinfection between the progeny of a virus could help to maintain viral population fitness. Conversely, the same SIE kinetics strongly limit opportunities for genetic interactions between the progeny of different viruses that have established distinct foci of infection. This would also be likely to have effects on viral fitness, for example, by preventing the unrestricted spread of viruses carrying defective interfering RNAs [48].

Importantly, as this patterning effect is due to the delayed nature of SIE rather than to a specific mechanism, we predict that it would be generalisable to the large number of other viruses

that establish SIE gradually during the infection of a cell, propagate locally within a host, and whose fitness and evolution are shaped by genetic exchange between viruses during coinfection [20,21,26]. We suggest that the spatial context of SIE is an important factor when considering viral diversity and fitness within hosts.

Although many viruses benefit from coinfection and genetic exchange, these processes have been studied particularly intensively for IAVs, in part because of the importance of reassortment in generating new pandemic strains through "antigenic shift." In addition to the widespread occurrence of reassortment in the natural evolution of IAV, a large body of evidence demonstrates reassortment of IAV during experimental coinfections both in vivo and in vitro [3,49]. Superficially, these observations might be seen as conflicting with our own data, which show that SIE should impose severe restrictions on coinfection between viruses that propagate locally within a host. However, there are several plausible explanations for why reassortant IAVs are regularly detected despite the effects of within-host SIE. Firstly, when considering epidemiological evidence for reassortment, the number of host organisms infected by IAVs is extremely large [50,51], providing ample opportunities even for rare interactions between viral strains within a host. Secondly, experimental studies of reassortment and coinfection within animals are often designed using high concentrations of viruses administered by artificial routes such as intranasal inoculation of mammals or injection into embryonated chicken eggs [39,52,53]. The delivery of a high concentration of viruses in a small window of time to the same anatomical site would be expected to increase the likelihood of coinfection of cells during an initial infection, when compared to natural infections. Thirdly, it is important to stress that although the timing of SIE greatly limits the number of infected cells in which coinfection is possible, it still allows a small proportion of cells to be coinfected. The reassortant progeny these cells produce could, if they have a fitness advantage, be rapidly amplified through replicating in other cells. For this reason, our results are compatible with studies that have observed reassortment of IAV in cases where simultaneous coinfection was not forced, for example, when the viruses were delivered in separate inoculation events or where the inoculation dose was relatively low [8,10,11].

We note that our study was designed to measure coinfection rather than reassortment. The chance of a "successful" reassortment depends on many different factors, such as segment mismatch, compatibility of packaging signals, and synchronicity of the viral lifecycles [54,55] and to exclude these effects, we chose to use a pair of isogenic viruses whose reassortment would not generate a novel genotype. Further studies will need to be performed to assess under what circumstances the small area of coinfection we observe are sufficient to support robust reassortment. We note that any factors that influence SIE, altering the proportion of cells where coinfection can occur, are likely to have a disproportionate impact on the likelihood of rare or unfavoured reassortment events, such as reassortants between viruses that are adapted to different host species.

Our findings do not identify a specific mechanism of SIE for IAVs but they do suggest possible mechanisms that may be relevant. We confirmed that SIE restricts coinfection after 6 h of primary infection [27,28,56] and went on to define its timing in more detail, demonstrating that SIE is not effective for the first 2 h of infection but then becomes increasingly effective in a way that suggests the presence of an exponentially increasing inhibitory factor. The timings of this would be consistent with the exponential accumulation of viral products from the primary infecting virus [57], consistent with previous observations indicating a connection between SIE and the presence of replicating influenza polymerase complexes [28]. This exponential inhibition model is also consistent with previous work showing that more coinfected cells could be detected if fewer replication-competent genome segments were delivered during primary infection [28], and with our own data showing that the amount of primary infecting

genomes determines the kinetics of SIE onset, as exponential relationships are sensitive to their starting parameters. Our data would be equally compatible with a directly acting viral factor or with a response by the host cell to the accumulation of viral factors. Interestingly, it seems from previous work that the interferon cascade is not involved in onset of SIE, as interferon competent and incompetent cell lines are equally able to induce SIE [28]. More work is required to determine if the mechanism of IAV SIE is due directly to the accumulation of products of replicating polymerases (either RNA transcripts or, indirectly, viral proteins) to a host-encoded factor that is produced in response to polymerase activity or to a combination of effects.

We propose here that spatial dynamics of coinfections are an underappreciated aspect of within-host evolution. We demonstrate that the timing of SIE creates a pattern of distinct infected lesions within host tissue that can directly impact the ability of viruses to undergo genetic exchange. Because of the impact of SIE on the local spread of viruses, increasing our understanding of the spatial dynamics of coinfections will help us to better understand the opportunities viruses have to interact and evolve within their hosts.

## Materials and methods

### Cells and viruses

MDCK cells (a gift from Prof. P Digard at the Roslin Institute, University of Edinburgh) and human embryonic kidney (HEK) 293T cells (a gift from Prof. S Wilson, MRC-University of Glasgow Centre for Virus Research) were maintained in complete media (Dulbecco's Modified Eagle Medium (DMEM, Gibco) supplemented with 10% foetal bovine serum (FBS, Gibco)). All cells were maintained at 37°C and 5% $CO_2$ in a humidified incubator.

The wild-type (WT) PR8 was generated in HEK293T cells using the pDUAL reverse genetics system, a gift of Prof. Ron Fouchier (Erasmus MC Rotterdam), as previously described [58]. ColorFlu viruses (ColorFlu-eGFP and ColorFlu-mCherry) were rescued in HEK293T cells from plasmids encoding the NS segment supplied by Prof. Y. Kaowaoka (University of Wisconsin-Madison, University of Tokyo), in addition to WT PR8 pDUAL plasmids edited to contain compensatory mutations (HA T380A and PB2 E712D, as previously described [39]). The viruses were passaged at low MOI in viral growth media (VGM) (DMEM with 0.14% (w/v) bovine serum albumin (BSA) and 1 μg/μl TPCK-treated trypsin) to create a working stock.

Virus plaque titres in PFUs per ml (PFU/ml) were obtained in MDCK cells under agarose, following the procedure of Gaush and Smith [59].

### Mouse infections

C57BL/6 mice (Charles River, United Kingdom) were infected intranasally with a total of 1,000 PFU of ColorFlu viruses (an equal mixture of mCherry and eGFP variants). All animal work was carried out in line with the EU Directive 2010/63/eu and Animal (Scientific Procedures) Act 1986, under a project licence P72BA642F, and was approved by the University of Glasgow Animal Welfare and Ethics Review Board. Animals were housed in a barriered facility proactive in environmental enrichment.

### Immunofluorescence and imaging

Confocal images of infected cells were obtained by infecting cells on coverslips, with an MOI of 0.5 PFU/ml for each of the ColorFlu viruses, for 8 h before fixation in 4% (v/v) formaldehyde diluted in PBS (Sigma). Following fixation, the cells were rinsed in PBS and the nucleus stained with 4′,6-diamidino-2-phenylindole (DAPI, Thermo Fisher). Coverslips were then

mounted and imaged with the Zeiss Laser Scanning 710 confocal microscope, images were processed using Zeiss Zen 2011 software.

To obtain images of viruses spreading from a coinfected focus, MDCK cell monolayers were infected with mCherry and eGFP tagged viruses both at an MOI of 5 PFU/ml. At 1 h p.i., the infected cells were dispersed with TrypLE express for 15 min (Thermo Fisher) and diluted in VGM to create a suspension that was applied to fresh MDCK cell monolayers. The cells were left to settle for 4 h, after which an agarose overlay was added and infections were left to proceed, as in a standard plaque assay.

To obtain images of interactions between initially separate foci of infection, MDCK cell monolayers were infected with a diluted mixture of mCherry and eGFP tagged ColorFlu viruses after which an agarose overlay was applied and infections were left to proceed as in a standard plaque assay. The infected plates were imaged through the agarose every 24 h in a Celigo imaging cytometer (Nexcelom). Images were processed in FIJI (ImageJ) [60] using custom macros that can be accessed at https://github.com/annasimsbiol/colorflu.

To obtain live cell images of spreading infections, ColorFlu-eGFP and mCherry viruses were diluted in VGM to an MOI of 0.5 (PFU/cell) and applied to confluent MDCK cell monolayers. Following a 1 h incubation, the inoculum was removed and agarose was overlaid, as in a standard plaque assay procedure [1]. The plate was transferred to an Observer Z1 live-cell imaging microscope (Zeiss, United States of America), and a tile from a well was imaged every 15 min over 72 h. The acquired videos were compiled using Zen (Zeiss).

To obtain images of infections in mice, at the indicated number of days, post infection animals were sacrificed and their lungs inflated with 2% low melt agarose. Lungs were then fixed in PLP buffer (0.075 M lysine, 0.37 M sodium phosphate (pH 7.2), 2% formaldehyde, and 0.01 M NaIO$_4$) overnight, 300 μm sections of lung were cut using a vibrotome, and imaging was performed using an LSM880 confocal microscope (Zeiss) using a 20× objective at 0.6× digital zoom with 5 μm z steps. Images were stitched and maximum intensity projections were made using Imaris software (version 9.7.0, Bitplane, USA).

## Viral growth kinetics

For single cycle growth kinetics, viruses were applied to confluent MDCK monolayers at an MOI of 2.5 PFU/cell, and the cells were incubated with the inoculum for 1 h at 37˚C and 5% CO$_2$ in a humidified incubator to allow the viruses to enter cells. Following this, the inoculum was removed, and the cells bathed in acid wash (10 mM HCl and 150 mM NaCl in MiliQ-water (pH 3)) for 1 min after which fresh VGM was added. Media were sampled at the time points indicated, clarified by low-speed centrifugation and stored at −80˚C before titration by plaque assay.

Multicycle kinetics were determined as above, except that the cells were infected at an MOI of 0.001 PFU/cell and the acid wash step was omitted.

## Flow cytometry

MDCK cells were inoculated for 1 h with ColorFlu-eGFP viruses diluted in VGM at the MOI indicated. After 1 h, the inoculum was removed and replaced with complete media. After the time intervals indicated, cells were inoculated for 1 h with Colorflu-mCherry, at the MOI indicated. After 1 h, the inoculum was removed and replaced with complete media, and the cells were incubated for a further 16 h at 37˚C. The proportions of cells expressing the different fluorophores were assessed using a Guava easyCyte HT System cytometer (Luminex). Briefly, infected and mock-infected MDCK monolayers were dissociated TrypLE express for 15 min (Thermo Fisher) and dispersed into a single-cell suspension before fixation in 2%

formaldehyde (v/v) in PBS. Each sample was prepared in technical triplicate and the data were analysed in FlowJo software v10.6. The thresholds for assessing positive detection of either red or green fluorophore were set using the mock-infected cells as a negative control. We noted that these experiments were technically challenging to repeat consistently, potentially due to the variable expression of fluorophores by different stocks of reporter virus. To ensure that comparable results were collected, for each experimental repeat, the volume of virus used was empirically adjusted so that, during simultaneous infection, the expression of the 2 reporter viruses was equivalent across all repeats.

## Modelling

The ability of viruses to infect cells was quantified by calculating how many reporter viruses could cause cells to express a fluorophore. This was done by adapting standard calculations for MOI, in which the proportion of cells infected with different numbers of viruses follows a Poisson distribution [61]. This makes the assumptions that (i) viruses that are added to cells at the same time will infect independently of each other; and (ii) at the point that the primary virus is added, all cells are equally permissive to infection. The concentration of viruses that could make cells express a fluorescent reporter can then be calculated from the proportion of cells that express the reporter, either alone or in combination with another reporter [58]. For example, the concentration per cell of "red forming units" (RFU; viruses that cause expression of the red fluorophore) is:

$$MOI_{RFU} = -\ln(1 - (P_R + P_{GR})),$$

where $MOI_{RFU}$ is the concentration of RFU per cell, $P_R$ is the proportion of cells that express only the red fluorophore, and $P_{GR}$ is the proportion of cells that express both red and green fluorophores.

The change in RFU per cell with increasing intervals between primary (green) and secondary (red) infection was modelled by assuming that during an initial interval there would be no SIE, after which point SIE would increase exponentially:

$$MOI_{RFU} = IF(t < t_S, MOI_{RFU}(0), MOI_{RFU}(0).e^{-K(t-t_s)}),$$

where $t$ is the interval between the primary and secondary infections, $t_s$ is the interval after which SIE becomes effective, $MOI_{RFU}(0)$ is the concentration of RFU per cell when the red and green viruses are added at the same time, and $K$ is the rate constant. Under this model, once SIE has begun, the concentration of RFU declines with a half-life of $ln(2)/K$. The model was fitted as a one phase decay curve by least squares, using GraphPad Prism (version 9.4.1; GraphPad). The initial fit of the model was carried out with the constraints that $K > 0\ h^{-1}$ and that as $t$ increased $MOI_{RFU}$ tended to 0 RFU/cell. In subsequent experiments when the ratios of red and green viruses were varied, the value of $t_s$ was constrained at 2 h.

## Supporting information

**S1 Fig. ColorFlu viruses tagged with mCherry and eGFP have similar growth kinetics. (A)** Single cycle growth kinetics of ColorFlu viruses were assessed by infecting MDCK cell monolayers at an MOI of 2.5 PFU/cell and harvesting the supernatant at the time points indicated. Virus titre was assessed using plaque assay on MDCK cells. **(B)** Multi-cycle growth kinetics of ColorFlu viruses were assessed by infecting MDCK cell monolayers at an MOI of 0.001 PFU/cell and harvesting the supernatant at the time points indicated. The mean and SD are shown ($n$ = 3). For all time points in A and B, the titres of ColorFlu-mCherry and ColorFlu-eGFP were not significantly different (Mann–Whitney U test, $p > 0.05$). LLOQ = Lower limit of

quantification. Underlying data can be accessed at the following address: http://dx.doi.org/10.5525/gla.researchdata.1370.
(TIF)

**S2 Fig. Cell death and fluorophore colours are considerations in the kinetics of SIE. (A)** Percentage of cells that were negative for both mCherry and eGFP expression, determined by flow cytometry. MDCK cells were infected with Colorflu-eGFP before secondary infection at the time points indicated with ColorFlu-mCherry, with both viruses at MOI 1 FFU/cell. Data presented as mean and SD (*n* = 6). MDCK cells were infected with either **(B)** Colorflu-eGFP or **(C)** ColorFlu-mCherry before secondary infection at the time points indicated with the other virus, with both viruses at MOI 1 FFU/cell. The percentage of fluorescent cells was then assessed using flow cytometry. The number of red and green forming units per cell (RFU, GFU) was calculated from the percentage of red, green, and coinfected cells under the assumption that infection follows a Poisson distribution. The number of secondary viruses detected per cell were used to fit a model in which the number of secondary viruses per cell that could be detected was constant for 2 h and then decayed exponentially to zero with increasing intervals between infections. The SST for the models in (B) and (C) are 0.22 and 0.24, respectively. Data are presented as mean and SD (*n* = 3). Underlying data can be accessed at the following address: http://dx.doi.org/10.5525/gla.researchdata.1370.
(TIF)

**S3 Fig. Superinfection exclusion limiting coinfection between distinct virus populations in vitro.** Viruses were seeded onto monolayers of MDCK cells, overlayed with agarose, and imaged every 24 h. Images taken on Celigo fluorescent microscope. Scale bar = 2 mm. Underlying data can be accessed at the following address: http://dx.doi.org/10.5525/gla.researchdata.1370.
(TIF)

**S4 Fig. Mouse infection with a mixture of influenza viruses, showing early foci of infection in the bronchi.** C57BL/6 mice were intranasally inoculated with mixtures of mCherry and eGFP expressing ColorFlu viruses (500 PFU of each virus). Lung sections, taken at 3 dpi, were imaged using a Zeiss LSM 800 with a 20× objective lens. Scale bar = 1,500 μm. Underlying data can be accessed at the following address: http://dx.doi.org/10.5525/gla.researchdata.1370.
(TIF)

**S5 Fig. Superinfection exclusion limits coinfection between distinct virus populations in vivo.** C57BL/6 mice were intranasally inoculated with mixtures of mCherry and eGFP expressing viruses (500 PFU of each virus). Lung sections, taken at 6 dpi, were imaged with a Zeiss LSM 800 using a 20× objective lens. **(A)** Confocal micrographs of whole lung slices from infected mice 6 dpi. **(A)** Whole lung images. Scale bar = 1,500 μm. **(B)** Enlarged images of infected lesions. Scale bar = 100 μm. Underlying data can be accessed at the following address: http://dx.doi.org/10.5525/gla.researchdata.1370.
(TIF)

**S1 Movie. Infected cells migrate towards the centre of influenza A virus plaques.** Red and green ColorFlu viruses were used to infect MDCK cells under agarose and observed over 72 h in Zeiss Livecell observer microscope using a 20× objective lens. Underlying data can be accessed at the following address: http://dx.doi.org/10.5525/gla.researchdata.1370.
(AVI)

## Author Contributions

**Conceptualization:** Anna Sims, Laura Burgess Tornaletti, Seema Jasim, Jack C. Hirst, Joanna Wojtus, Elizabeth Sloan, Luke Thorley, Edward Hutchinson.

**Funding acquisition:** Chris Boutell, Edward Roberts, Edward Hutchinson.

**Investigation:** Anna Sims, Laura Burgess Tornaletti, Seema Jasim, Chiara Pirillo, Ryan Devlin, Edward Roberts.

**Methodology:** Anna Sims, Laura Burgess Tornaletti, Seema Jasim, Chiara Pirillo, Ryan Devlin, Jack C. Hirst, Colin Loney, Joanna Wojtus, Elizabeth Sloan, Luke Thorley, Edward Roberts, Edward Hutchinson.

**Supervision:** Chris Boutell, Edward Hutchinson.

**Visualization:** Anna Sims, Laura Burgess Tornaletti, Seema Jasim, Jack C. Hirst, Colin Loney, Edward Roberts.

**Writing – original draft:** Anna Sims.

**Writing – review & editing:** Anna Sims, Edward Hutchinson.

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
