## [Editor Report · Decision Letter 0]

7 Jun 2022

Dear Dr. Hutchinson, 

Thank you for submitting your manuscript entitled "Superinfection exclusion creates spatially distinct influenza virus populations" for consideration as a Short Reports by PLOS Biology.

Your manuscript has now been evaluated by the PLOS Biology editorial staff and I am writing to let you know that we would like to send your submission out for external peer review. After discussing with the team, we think that this manuscript will fit better the Short Report Article type. Please select Short Report where corresponds when doing the full submission in the system. 

Before we can send your manuscript to reviewers, we need you to complete your submission by providing the metadata that is required for full assessment. To this end, please login to Editorial Manager where you will find the paper in the 'Submissions Needing Revisions' folder on your homepage. Please click 'Revise Submission' from the Action Links and complete all additional questions in the submission questionnaire.

Once your full submission is complete, your paper will undergo a series of checks in preparation for peer review. After your manuscript has passed the checks it will be sent out for review. To provide the metadata for your submission, please Login to Editorial Manager (https://www.editorialmanager.com/pbiology) within two working days, i.e. by Jun 09 2022 11:59PM.

Kind regards,

Paula

Senior Editor

PLOS Biology

---

## [Decision Letter · Decision Letter 1]

11 Jul 2022

Dear Dr Hutchinson,

Thank you for your patience while your manuscript "Superinfection exclusion creates spatially distinct influenza virus populations" was peer-reviewed at PLOS Biology. It has now been evaluated by the PLOS Biology editors, an Academic Editor with relevant expertise, and by several independent reviewers. 

In light of the reviews, which you will find at the end of this email, we would like to invite you to revise the work to thoroughly address the reviewers' reports.

After discussion with the Academic Editor, we consider that it is important that you pay particular attention to the framing and focus, such that the novelty is more apparent to non-influenza readers. Specific comments from the reviewers will be helpful in this process. Regarding the issues raised by reviewer #1 and #2, it would be important to expand the evidence about the timing of SIE, specifically why the authors assume it starts before 4 hours, being important to add more time points between 2 and 6 hrs to ensure the curve truly matches. Additionally, how the RFU and GFU are calculated needs to be further explained, clarifying whether this supposed to be red fluorescent units and green fluorescent units and measuring the intensity of fluorescence in each cell. Other points from reviewer #2 can be addressed with refinements in the text such as more explanation of concepts or even removal of some text that the reviewers note appear tangential or distracting. Please, also address experimentally the issues raised by reviewers #1 and #3 regarding your experimental approach and controls. Finally, it is also important to address the issue by reviewer #1 regarding whether eGFP and mCherry viruses have similar fitness. This can be addressed either by competition assay or controlling for potential issues by reversing the order of viruses in key experiments (e.g. mCherry then eGFP or eGFP then mCherry) to make sure these apparent growth differences don't affect the results and interpretation. competition assay or controlling for potential issues by reversing the order of viruses in key experiments (e.g. mCherry then eGFP or eGFP then mCherry) to make sure these apparent growth differences don't affect the results and interpretation. For example, in Figure 1B, mCherry single infected cells seem to be much greater than eGFP single infected cells when introduced together. Flipping the order in this experiment would be helpful.

Given the extent of revision needed, we cannot make a decision about publication until we have seen the revised manuscript and your response to the reviewers' comments. Your revised manuscript is likely to be sent for further evaluation by all or a subset of the reviewers.

**IMPORTANT - SUBMITTING YOUR REVISION**

*Re-submission Checklist*

*Published Peer Review*

*PLOS Data Policy*

*Blot and Gel Data Policy*

Sincerely,

Paula

---

Senior Editor

PLOS Biology

REVIEWS:

Reviewer #1: Genetics and Evolution of RNA Viruses.

Reviewer #2: Virus-Host interactions.

Reviewer #3: Anice C. Lowen. IAV evolutionary dynamics, reassortment and recombination.

Reviewer #1: In this manuscript, Sims and colleagues use a clever experimental design with two isogenic IAV that express distinct fluorescent reports to examine the phenomenon of superinfection interference. This is not the first treatment of the topic (see cited works of Sun and Brooks, Dou et al. for example). Those two prior works suggested that superinfection was mediated by viral replication, which the authors refer to and which is consistent with the findings of the present study. The time frame for superinfection (and viral replication) is similar in the present study as well, although the authors narrow down the window in this experiment, while providing further evidence for the exponential model of superinfection exclusion kinetics.

The main new finding of this manuscript is a visualization of how superinfection inclusion acts to limit co-infection with viruses from distinct foci in vitro and in vivo. This spatial isolation of infectious foci has implications for understanding the limits to reassortment in vivo, an important issue in influenza genetics. The data are well presented and the discussion is quite clear. I only have a couple specific quibbles with the data as presented.

Overall, this is a nice addition to a rapidly growing literature on influenza dynamics in culture and in vivo, highlighting spatial and temporal aspects of coinfection, superinfection, not to mention defective interfering and semi-infectious particles. It will be of interest to the influenza community, and virologists in general.

Major points/suggestions

It isn't clear from the data presented that the eGFP and mCherry viruses have similar fitness. One step growth curves are not particularly sensitive (Supp Data 1) and it appears that they do have differences in single cycle. Suggest either a competition assay or controlling for potential issues by reversing the order of viruses in key experiments (e.g. mCherry then eGFP or eGFP then mCherry) to make sure these apparent growth differences don't affect the results and interpretation. For example, in Figure 1B, mCherry single infected cells seem to be much greater than eGFP single infected cells when introduced together. Flipping the order in this experiment would be helpful.

Although I am inclined to agree that viral replication underlies the mechanism of SIE, wouldn't there be other mechanisms that could lead to exponential changes other than viral replication itself? (For example cell intrinsic immune cascades? I do realize that Sun and Brooke found it appeared to be interferon independent)

Minor issues

Supp Data 1 - In single cycle growth curve, which is arguably more important here given time frames employed in experiments, the mCherry virus does appear to have an advantage over the eGFP one up to 30 hours. It does not appear that statistics were done on panel A, only B? Also, number of replicates should be specified (e.g. points per time point). Unclear why MWU test used rather than t-test (are they non-normally distributed)? This has relevance to assessment of relative fitness of the viruses.

Line 63 - Suggest removing "therefore" from this sentence (used to two successive sentences)

Line 199 - should read Figure 2BC?

Line 203 - should read Figure 2B?

LIne 350 - should read "could be" instead of "could are" or simply "are"

Reviewer #2: The main points of this manuscript are that significant SIE can be seen between 4-6 hrs, SIE prevents plaques from being co infected unless cells are infected by both viruses around a similar time, and that SIE also occurs in the lungs of mice. All these conclusions confirm work in previous studies, as acknowledged by the authors, and it's unclear what is novel about the work presented in this paper in addition to using colored viruses. A better distinction of what is previously known and what this paper is adding to the field is critical. Additionally, the authors bring up multiple theories and topics such as genomic reassortment, DI-RNAs, semi-infectious particles, etc. but none of these are studied in this work. The impact of this paper would greatly improve if the authors could address any of these topics in their experiments. By including statements about these topics, the authors present gaps that their experiment are not addressing. Overall, the authors need to clarify what is novel about this paper versus previous works and add additional experiments addressing forms of interference that could be pivotal in SIE. 

Major comments:

* The authors comment on genome interactions throughout the manuscript, but do not show any evidence of genomic interaction in any of their experiments. The use of isogenic virus maintains replication rates of the 2 viruses but prevents the study of reassortment and rearrangement of viral genomes. To enhance this paper, the authors should generate viruses with trackable mutations that do not alter replication rates or capabilities but would allow for detection of genomic interactions and reassortments. 

* In Figure 1, the authors state that SIE begins between 2 and 4 hrs post initial infection, but the evidence for this need to be expanded. Currently, the SIE at 4 hrs does not differ from 0 hrs between infections, so why do the authors assume SIE begins before then? The mathematical curve does show a beginning of a dip for RFU cells, but more timepoints particular between 2 and 6 hrs are needed to ensure the curve truly matches. Additionally, how the RFU and GFU are calculated needs to be further explained. Is this supposed to be red fluorescent units and green fluorescent units and measuring the intensity of fluorescence in each cell?

* Figures 1 and 2 appear to confirm previous reports that 6 hrs between infections is the first timepoint SIE is observed without contributing new knowledge. Additionally, having more cells infected with the primary virus prevents the secondary virus about 30 min faster is what appears to be the only novel data here. Also, in the figure legend, it is stated that these experiments are done at an MOI of 0.5. From the images, it seems that more than 75% of the cells are infected suggesting a higher MOI.

* Line 348, How does the capability for reassortment contrast with your data? Data show that coinfection can occur during SIE conditions in cell culture and mice albeit at low percentages of cells. How many cells do you expect you would need for reassortment to occur? I would expect a few coinfected cells would be sufficient. Since isogenic viruses are used in this manuscript, there is no data on reassortment, and the discussion should acknowledge this fact.

* Line 364-367, there are no data in this manuscript regarding coinfection using different strains. 

* I am confused why DI-RNAs are brought into the discussion. If a DI-RNA is present in a lesion, wouldn't that inhibit the viral replication in that lesion and prevent spread of that virus to another lesion? Data and experiments shown do not address DI-RNAs, so why are these in the introduction and discussion?

Minor comments:

* Line 52, DI-RNAs can be more than large internal deletions. Insertions, copy-backs, snapbacks, etc. can also produce DI RNAs. Even a 1 nt deletion can produce a frameshift that knocks out an essential protein and produces a DI RNA. 

* Is supplementary figure 1 A+B truly single cycle vs multi cycle? Can the authors comment what's the rationale of these experiments if the goal is to compare growth kinetics. 

* Figure legend for 1A, better to state size of bar in pictures (1 uM?) and not the 64x objective. 

* Fig. 2A For ease of reading, please write the ratios of Green to Read virus instead of 

Reviewer #3: Sims et al. report a series of elegant experiments designed to interrogate the impact of super-infection exclusion (SIE) on the spatio-temporal dynamics of influenza A virus infection. Main findings include that SIE initiates around 4 h post-infection in their system; SIE does not block coinfection within an expanding viral focus; but SIE does block coinfection between independent foci that expand to occupy the same space. These dynamics were observed both in cell culture and in a mouse model. 

The work is well thought-out and rigorously performed. Although SIE was first observed decades ago, the work offers important novel insight into the effects of SIE on within-host spatial dynamics and will be of broad interest. The findings are likely to be relevant to other respiratory viruses. My comments below mainly relate to the need for refinement of interpretations of the data and more in-depth thinking on how the results reported here relate to the published literature. 

1. The data don't appear to support the conclusion that there is "a progressive shift from a permissive to exclusionary state beginning around 2h post-primary infection" (Lines 168 and 381). Rather, the first statistically significant evidence of exclusion is seen at 6h and no exclusion is apparent at 2h. 

2. Fig. 2: The measured RFU/cell declines with increasing dose of primary virus. This effect is seen consistently across replicates and time points but is not discussed. At the high MOIs used, limitations on cellular resources may result in interference within coinfected cells.

3. Line 125: The description of SIE as rapid is subjective. Four hours is a lot of time in the life of an influenza virus.

4. Figure 2C: About 50% of cells are positive for green, but red is excluded from about 90% of cells. What might explain this?

5. Discussion of how to reconcile SIE with data on reassortment in experimental systems overlooks a number of relevant published observations: 

- reassortment is common even when coinfecting viruses are delivered through two independent transmission events (Tao et al. JVI 2015) 

- reassortment is common and increases over time even when relatively low inoculation doses are used (e.g. 1000 PFU; Tao et al. JVI 2014)

- reassortment is common when co-infecting viruses are delivered to guinea pigs 12 h apart (Marshall et al. PPath 2013)

Better explanations for abundant reassortment in the context of SIE include: 

- the difference in tissue sites examined (nasal for reassortment; lung in the present work)

- The difference in virus strains used. PR8 replicates more quickly than most IAVs and is likely to induce SIE more rapidly. Multiple have been tested for reassortment and reassortment frequencies vary (Phipps et al. NatMicro 2020). 

- differences in host species may affect kinetics of innate responses and spatial dynamics of viral dispersal

Minor comments:

1. Line 2: check short title - perhaps a word is missing.

2. Line 66-67: Rephrase for clarity. 'Viral genomics' is vague and 'imply' does not convey well the absolute requirement of coinfection for reassortment. Perhaps "Detection of reassortant viruses in circulation indicates that cellular coinfection must occur with some frequency."

3. Line 72: Note that the frequency of coinfection would not by modulated by selection. This experiment shows that the frequency of cellular coinfection in this model host is sufficiently high to sustain viral replication over several days.

4. Fig 1A: please define the scale bar and add units to the MOI given in the figure legend.

5. Line 190: I suggest rephrasing this subtitle in light of the fact that only two parameters were tested - so a claim that the main determinant was identified seems to overstate things.

6. Line 195 and throughout: the effort to handle the data quantitatively is commendable, but IC50 doesn't seem to be the right term here in that the time interval leading to 50% inhibition, rather than a concentration, is estimated.

7. Line 199: Figure 3 is referred to erroneously.

8. Line 233: The reference to a 'switch' here suggests a binary state and may want to be reconsidered. The inability to overcome SIE with a 5-fold increase in the dose of the second virus makes sense if SIE is driven by a factor that accumulates exponentially (as suggested by the authors).

9. Line 285: "In a natural infection we assume that it is unlikely that multiple 'incoming' viruses would reach the same cell within a short space of time." This statement should be revised in light of the fact that essentially simultaneous co-infection would occur commonly as viruses budding out of one cell infect a neighboring cell.

---

## [Editor Report · Decision Letter 2]

3 Nov 2022

Dear Dr. Hutchinson,

Thank you for your patience while we considered your revised manuscript "Superinfection exclusion creates spatially distinct influenza virus populations" for publication as a Short Reports at PLOS Biology. This revised version of your manuscript has been evaluated by the PLOS Biology editors and the Academic Editor.

Based on our Academic Editor's assessment of your revision, we are likely to accept this manuscript for publication, provided you satisfactorily address the following data and other policy-related requests.

1. DATA POLICY:

You may be aware of the PLOS Data Policy, which requires that **all data be made available without restriction**: http://journals.plos.org/plosbiology/s/data-availability. For more information, please also see this editorial: http://dx.doi.org/10.1371/journal.pbio.1001797

A) Supplementary files (e.g., excel). Please ensure that all data files are uploaded as 'Supporting Information' and are invariably referred to (in the manuscript, figure legends, and the Description field when uploading your files) using the following format verbatim: S1 Data, S2 Data, etc. Multiple panels of a single or even several figures can be included as multiple sheets in one excel file that is saved using exactly the following convention: S1_Data.xlsx (using an underscore).

B) Deposition in a publicly available repository. Please also provide the accession code or a reviewer link so that we may view your data before publication.

Regardless of the method selected, please ensure that you provide the individual numerical values that underlie the summary data displayed in the following figure panels as they are essential for readers to assess your analysis and to reproduce it: Figures 1CDE, 2ABC, 3E, 4C, and Supplementary Figures S1AB, S2ABC.

**Please also ensure that figure legends in your manuscript include information on where the underlying data can be found, and ensure your supplemental data file/s has a legend.**

**Please ensure that your Data Statement in the submission system accurately describes where your data can be found.**

2. Please add size bars to the microscopy figures 3D and 4BE.

We expect to receive your revised manuscript within two weeks.

*Published Peer Review History*

*Press*

Sincerely,

Paula

---

Senior Editor,

pjaureguionieva@plos.org,

PLOS Biology

---

## [Editor Report · Decision Letter 3]

2 Dec 2022

Dear Ed,

Thank you for the submission of your revised Short Reports "Superinfection exclusion creates spatially distinct influenza virus populations" for publication in PLOS Biology. On behalf of my colleagues and the Academic Editor, Andrew Mehle, I am pleased to say that we can in principle accept your manuscript for publication, provided you address any remaining formatting and reporting issues. These will be detailed in an email you should receive within 2-3 business days from our colleagues in the journal operations team; no action is required from you until then. Please note that we will not be able to formally accept your manuscript and schedule it for publication until you have completed any requested changes.

PRESS

Sincerely, 

Paula

---

Senior Editor

PLOS Biology
